# Clinical Exome Sequencing Revealed a De Novo *FLNC* Mutation in a Child with Restrictive Cardiomyopathy

**Francesca Girolami** [1,*,†] **, Silvia Passantino** [1,†] **, Adelaide Ballerini** [1] **, Alessia Gozzini** [1] **, Giulio Porcedda** [1] **, Iacopo Olivotto** [1,2] **and Silvia Favilli** [1]

1   Cardiology Unit, Meyer Children's Hospital, 50139 Florence, Italy; silvia.passantino@meyer.it (S.P.); adelaide.ballerini@stud.unifi.it (A.B.); alessia.gozzini@meyer.it (A.G.); giulio.porcedda@meyer.it (G.P.); iacopo.olivotto@unifi.it (I.O.); silvia.favilli@meyer.it (S.F.)
2   Cardiomyopathy Unit, Careggi University Hospital, University of Florence, 50134 Florence, Italy
*   Correspondence: francesca.girolami@meyer.it
†   These authors contributed equally to this work.

**Abstract:** Restrictive cardiomyopathy (RCM) is a rare disease of the myocardium caused by mutations in several genes including *TNNT2, DES, TNNI3, MYPN* and *FLNC*. Individuals affected by RCM often develop heart failure at a young age, requiring early heart transplantation. A 7-year-old patient was referred for genetic testing following a diagnosis of restrictive cardiomyopathy. Clinical exome sequencing analysis identified a likely pathogenic mutation in the *FLNC* gene [(NM_001458.5 c.6527_6547dup p.(Arg2176_2182dup)]. Its clinical relevance was augmented by the fact that this variant was absent in the parents and was thus interpreted as de novo. Genetic testing is a powerful tool to clarify the diagnosis, guide intervention strategies and enable cascade testing in patients with pediatric-onset RCM.

**Keywords:** restrictive cardiomyopathy; exome sequencing; *FLNC*

## 1. Introduction

Restrictive cardiomyopathy (RCM) is a rare myocardial condition characterized by a restrictive filling pattern and reduced diastolic volume of one or both ventricles; systolic function is usually normal until advanced stages of the disease, as is wall thickness—except in patients with infiltrative processes, in whom it may be increased [1,2]. RCM is the rarest form of cardiomyopathy (4.5% of the total) with an annual incidence of 0.03–0.04/100,000 children [3]. A significant subgroup of patients has mixed phenotypes, overlapping hypertrophic cardiomyopathy (HCM) [4]. RCM is associated with severe heart failure, limited response to treatment and frequent need for heart transplantation. Early symptoms associated with RCM may be non-specific, including fatigue and effort intolerance. Objective findings include elevated systemic and pulmonary venous pressure, hepatomegaly, pulmonary edema and pulmonary hypertension. LV ejection fraction is typically preserved although, in the final stages of the disease, systolic dysfunction may ensue. Due to the aggressive nature of this disease, patients are at risk of sudden death, sustained atrial arrhythmias, conduction disease and thromboembolism [5].

In pediatric RCM patients, survival 5 years after diagnosis is only 68% and is particularly low in non-transplanted children with pure RCM compared to those with a mixed phenotype. Other high-risk features include the presence of heart failure symptoms and small cavity dimensions. Marked left atrial dilatation and the need for diuretics during follow-up are similarly associated with higher mortality. No approved medical therapy exists for the treatment of diastolic dysfunction due to RCM; systemic venous and pulmonary congestion can be improved with the use of diuretics. Heart transplantation is associated with a 1- and 5-year survival rate of 89% and 77%, respectively; at 10 years post-transplant, the survival rate is comparable to children with other forms of cardiomyopathy [3,6]. An

increasing number of sarcomeric and non-sarcomeric mutations have been described in RCM, corroborating the concept of overlap with other forms of cardiomyopathy. When RCM is successfully genotyped, it is mostly related to variants in sarcomeric genes that also cause HCM, such as *TNNI3, TNNT2, MYH7* and *MYBPC3*. Variants in *TPM1, MYL2* and *MYL3* genes are found mainly in isolated forms; *ACTC1* and *TNNC1* variants have been observed in families with a mixed RCM/HCM phenotype [7–11]. In addition, variants in the *DES* gene, codifying desmin, are also associated with RCM, typically with conduction abnormalities including advanced atrioventricular block [12].

Recently the *FLNC* gene, coding for Filamin C, has been associated with RCM [13]. The prevalence of this gene in cardiomyopathies, the nature and localization of variants in the protein and its impact on diagnosis, prognosis and therapy, have not yet been well defined. FLNC variants causing different cardiomyopathies show genotype–phenotype correlations based on variant type and position in the protein domains. DCM-associated FLNC variants are usually truncating, presumably leading to haploinsufficiency, while HCM-associated variants are missense, i.e., acting by a dominant negative mechanism [14,15]. FLNC variants causing RCM may be missense, or rarely in-frame or frameshift insertion or deletion and are localized mainly in the ROD2 sub-domain [13] of the protein. Given its structural characteristics, it is assumed that Filamin C plays a role in sarcomeric mechano-transduction in muscle cells. The *FLNC* gene consists of 48 exons and is located on chromosome 7q32–35. Two different isoforms have been described: the longest one is less abundant in cardiomyocytes in basal conditions but increases rapidly following cardiac stress. In the Z disk, Filamin C interacts with numerous proteins partly linked to cardiomyopathies, such as Mypalladin which may be related to RCM [16–22].

The reported estimated prevalence of FLNC variants in RCM patients has been poorly defined, ranging from 2.2% [16] to 8% [13]. In the HGMD Professional database (2022.1), 156 disease-causing variants in FLNC are reported, associated with an isolated cardiomyopathy (HCM, DCM, RCM, LVNC) and with a myopathy phenotype; 11/156 FLNC variants (7%) were associated with RCM. Because of the rarity of RCM due to FLNC variants, its influence on prognosis and therapy is poorly defined compared with other cardiomyopathies, particularly in children [14].

FLNC missense variants can cause early-onset RCM associated with variable degrees of skeletal myofibrillar myopathy, combined with mild CK elevation and supraventricular arrhythmias. In contrast to missense mutations, FLNC truncating variants have been found in patients with an arrhythmogenic/DCM phenotype characterized by a high risk of life-threatening ventricular arrhythmias [15,16].

## 2. Case Report

We describe the case of a 7-year-old girl, (body weight: 23 kg, height: 126 cm, BSA 0.9mq). She was born to non-consanguineous parents, had normal growth and psychomotor development and had a medical history remarkable only because of frequent inflammation of the upper airways with wheezing due to rhinovirus infection and laryngitis on an allergic basis. During hospitalization for EBV infection, a suspicion of long QT was raised, for which she underwent cardiological evaluation. Her electrocardiogram showed normal sinus rhythm, bi-atrial enlargement and ST segment depression in the inferior leads. Her Qtc interval was in the normal range (Figure 1). Echocardiography showed bi-atrial enlargement (left atrial volume: 40 mL/mq, right atrial volume: 37 mL/mq), mild systolic dysfunction (LVEF 0.55) with normal LV diameter: 34/25 mm and Z-score 1.4/0.2 SF: 25%. The LV diastolic pattern was pseudo-normal and TDI values were reduced. Doppler investigation showed mild mitral and tricuspid regurgitation. She was diagnosed with primary RCM and started on diuretics (furosemide 25 mg /day). General conditions were good, with normal vital signs and laboratory parameters, including cardiac biomarkers (Troponin I and NT-proBNP) constantly in the normal range. Physical examination including cardiac auscultation was normal, and she had no evidence of JVP elevation, peripheral edema or hepatomegaly.

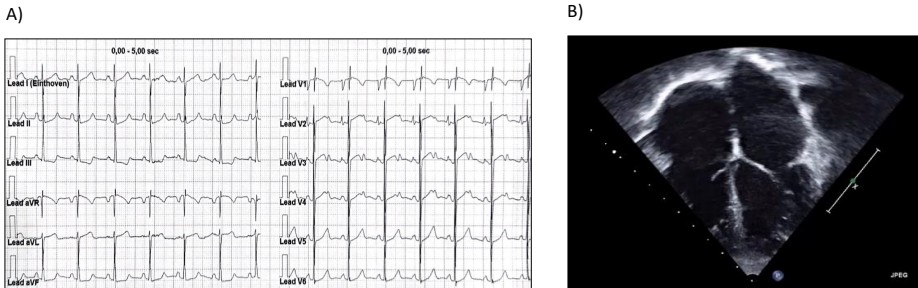

**Figure 1.** Proband's instrumental evaluation (EGG and echocardiogram). (**A**) ECG shows normal sinus rhythm, bi-atrial enlargement and ST segment depression in inferior leads. (**B**) Echocardiogram shows an RCM: bi-atrial enlargement and normal ventricular volume.

Due to elevated pulmonary vascular resistance at cardiac catheterization, the patient has been sent for an evaluation for possible inclusion on the heart transplant list.

Her family history was negative for cardiomyopathies or sudden death. Associated external malformations, facial dysmorphisms and signs of systemic involvement were excluded. Neurological examination was normal. A comprehensive metabolic screening was negative.

Following genetic counseling, Next Generation Sequencing (NGS) testing was performed with a panel of 174 genes using the TruSight™ Cardio Sequencing Kit (Illumina Inc., San Diego, CA, USA). No causative variants associated with the disease were identified. Thus, the NGS data were re-analyzed using clinical exome sequencing. Assessment of the clinical exome on the patient/parent trio identified the variant c.6527_6547dup p. (Arg2176_2182dup) in the *FLNC* gene, in heterozygosity (Ensembl transcript id ENST00000325888.8; Genebank transcript id NM_001458.5). Sanger sequencing was used to confirm the variant, which was not present in the DNA of the proband's parents and was adjudicated as de novo. It is an inframe duplication of 21 base pairs, not present in the database of allele frequencies GnomAD, and has not been previously described in patients. According to the ACMG guidelines [23], the variant classifies as likely pathogenic. Being a de novo variant, the younger sister of the proband was not tested (Figure 2).

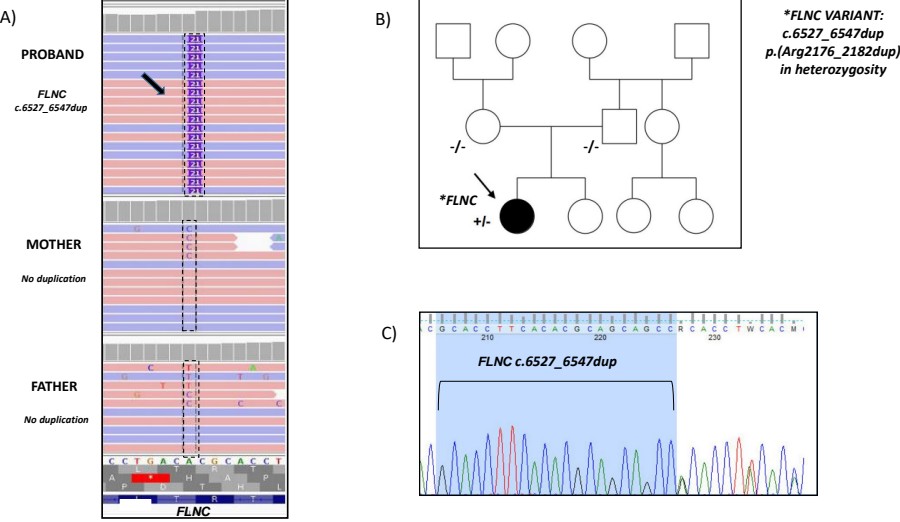

**Figure 2.** Pedigree and mutation analysis of the family. (**A**) Visualization by IGV software of NGS data (through alignment of enriched sequences to Hg 19) shows a 21-base pair duplication (purple) in the *FLNC* gene. (**B**) The proband (arrow) carried the variant c.6527_6547dup p. (Arg2176_2182dup) in heterozygosity in the *FLNC* gene. The variant was not found in the parents, resulting de novo. (**C**) The variant detected in the proband by NGS have been confirmed by Sanger sequencing.

## 3. Discussion

Variants in the *FLNC* gene have only recently been associated with pediatric-onset inherited cardiomyopathies, including RCM [24–26]. Thus, *FLNC* is not yet included in most commercial NGS panels for the identification of variants associated with cardiomyopathies and, in the patient described, identification of the pathogenic variant required clinical exome technology. The analysis of the clinical exome on the patient/parent trio made it possible to classify the variant as de novo. Recent data suggest that de novo variants in *FLNC* are related to earlier disease onset: in a recent report, three RCM patients with *FLNC* mutation all presented with the disease within the first three years of life [24]. In our patient, the variant c.6527_6547dup p.(Arg2176_2182dup) was identified in heterozygosity as an inframe duplication of 21 base pairs, not present in the GnomAD allele frequency database. In-frame deletions localized in the same region of the protein have recently been reported as causative of RCM [13]. A pediatric diagnosis of RCM-associated *FLNC* should be regarded as a high-risk condition, particularly in the presence of congestive symptoms and/or systolic dysfunction, warranting early consideration for heart transplantation. In addition, its de novo nature allowed reassurance of the parents and sister, eliminating the need for further follow-up and testing. As for other high-risk genes (such as those coding for lamin A/C, phospholamban, desmin and desmoplakin) a positive FLNC genetic test is critical in guiding clinical management. Furthermore, as a general rule, this case illustrates the importance of further testing in the presence of a negative NGS panel in children, moving forward to clinical exome sequencing: even though the yield may be low, any additional information may be of major clinical relevance [27,28].

## 4. Conclusions

Variants in the *FLNC* gene have recently been associated with pediatric-onset RCM. Recent data suggest that de novo variants in the *FLNC* gene are associated with high-risk status, earlier disease onset and adverse outcomes. We identified a de novo pathogenic variant using clinical exome technology. Personalizing genetic testing strategies may yield clinically important results in children with cardiomyopathies.

*Key Clinical Message*

Genetic investigation of pediatric cardiac restrictive phenotype by clinical exome sequencing is a powerful tool to clarify the diagnosis, identify the etiology and offer genetic counseling to the family.

**Author Contributions:** F.G. and S.P. contributed to the conception and design of the work. F.G. and S.P. made substantial contributions to the analysis and data interpretation and critically revised the manuscript for important intellectual content analyzed. F.G., S.P., A.B., A.G., G.P., I.O. and S.F. drafted the manuscript. All the authors critically revised the manuscript. All authors have read and agreed to the published version of the manuscript.

**Funding:** This research received no external funding.

**Institutional Review Board Statement:** The study was conducted in accordance with the Declaration of Helsinki and approved by the Institutional Review Board.

**Informed Consent Statement:** Informed consent was obtained from all subjects involved in the study.

**Data Availability Statement:** Not applicable.

**Acknowledgments:** We are thankful to all patients. We thank Davide Mei (Neuroscience Department, Children's Hospital A. Meyer-University of Florence, Italy) for the help in the bioinformatics analysis of Next Generation Sequencing data.

**Conflicts of Interest:** The authors declare no conflict of interest.

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
