# Peer review of "Clinical Exome Sequencing Revealed a De Novo FLNC Mutation in a Child with Restrictive Cardiomyopathy"

_cardiogenetics, doi:10.3390/cardiogenetics12020019_

Round 1

Reviewer 1 Report

In the manuscript titled “Clinical Exome Sequencing Revealed a de novo FLNC Mutation in a Child with Restrictive Cardiomyopathy”, the authors revealed a likely pathogenic mutation in FLNC gene [(NM_001458.5 c.6527_6547dup 15 p.(Arg2176_2182dup)] by Clinical exome sequencing analysis in a suspected isolated restrictive cardiomyopathy. The article is the first to report this mutation which is meaningful. I have the following suggestions to improve the manuscript:

  • As we know, previous literature reported early-onset restrictive cardiomyopathy (RCM) in congenital myopathy due to FLNC mutations. In the background description of “Introduction”, the authors mentioned this gene become popular in cardiomyopathies and it can be related to RCM. If they can provide more literature citations and describe how the mutation cases are unique in the diagnosis, prognosis, and therapy compared with other RCM cases, it will be more convincing.
  • About case description, the authors referred to “a medical history only remarkable frequent inflammation of the upper airways with wheezing.” The description of the medical history is too simplistic. The clinical symptoms are common symptoms of many respiratory diseases. If the authors can describe the causes, the duration and process of the onset in more detail may be more convincing. And RCM is often accompanied by a heart murmur, it may be more convincing if the authors can supplement the physical examination, especially cardiac auscultation and palpation.
  • Mutations in FLNC gene have been associated with human hypertrophic cardiomyopathy, restrictive cardiomyopathy, and a higher incidence of sudden cardiac death. At present, this mutation can only be an evidence to support the diagnosis, not the confirmation of the diagnosis. Regarding what the authors mention “The test is strongly recommended in patients with hereditary cardiomyopathies, in this way the test can also have a predictive role, in ascertaining which family members are most at risk.” Is it possible to provide evidence for this claim such as what is the incidence of RCM in the presence of FLNC mutations?

Author Response

 Reply to the evaluation by the Reviewer #1

  1. As we know, previous literature reported early-onset restrictive cardiomyopathy (RCM) in congenital myopathy due to FLNC mutations. In the background description of “Introduction”, the authors mentioned this gene become popular in cardiomyopathies and it can be related to RCM. If they can provide more literature citations and describe how the mutation cases are unique in the diagnosis, prognosis, and therapy compared with other RCM cases, it will be more convincing.

 Answer: Thank you for the comments. We add the following citations:  [14] Brodehl A., Gerull B., Genetic Insights into Primary Restrictive Cardiomyopathy. Review J Clin Med 2022 Apr 8;11(8):2094; [15] Ditaranto R., Caponetti A.G., Ferrara V., Parisi V.,Minnucci M.,Chiti C.,Baldassarre R., Di Nicola F.,Bonetti S.,Hasan T.,Potena L.,Galiè N.,Ragni L., Biagini E. Pediatric Restrictive Cardiomyopathies. Review Front Pediatr 2022 Jan 25;9:745365. 

In the text we add a brief paragraph to explain the genotype-phenotype correlations known to be related to FLNC gene  (Page 2 lines 56-63) “Recently the FLNC gene, coding for Filamin C, has been be associated with RCM [13]. The prevalence of this gene in cardiomyopathies, the nature and localization of variants in the protein and its impact on diagnosis, prognosis and therapy, have not yet been well defined. FLNC variants causing different cardiomyopathies show genotype-phenotype correlations based on variant type and position along the protein domains. DCM-associated FLNC variants are usually truncating, presumably leading to haploinsufficiency, while HCM-associated variants are missense, i.e. acting by a dominant negative mechanism [14-15]. FLNC variants causing RCM may be missense, or rarely in-frame or frameshift insertion or deletion and are localized mainly in ROD2 sub-domain [13] of the protein”.

In addition we add the sentences on page 4 lines 42-45: “A pediatric diagnosis of RCM associated FLNC should be regarded as a high-risk condition, particularly in the presence of congestive symptoms and/or systolic disfunction, warranting early consideration for heart transplantation” and the new reference [28] Girolami F., Iascone M., Pezzoli L., Passantino S., Limongelli G., Monda E., Rubino M., Adorisio R., Lombardi M., Ragni L., Olivotto I., Favilli S. Indicazioni all’esecuzione del test genetico nella diagnosi delle cardiomiopatie ad esordio pediatrico: percorso clinico della Società Italiana di Cardiologia Pediatrica. G Ital Cardiol 2022;23

  1. About case description, the authors referred to “a medical history only remarkable frequent inflammation of the upper airways with wheezing.” The description of the medical history is too simplistic. The clinical symptoms are common symptoms of many respiratory diseases. If the authors can describe the causes, the duration and process of the onset in more detail may be more convincing. And RCM is often accompanied by a heart murmur, it may be more convincing if the authors can supplement the physical examination, especially cardiac auscultation and palpation.

Answer: Thank you for the comments.  We added  in the text more details about the medical history and physical examination:

Page 2 lines 86-87 we add the sentences: “She was born from non-consanguineous parents, had normal growth and psychomotor development and a medical history remarkable only because of frequent inflammation of the upper airways with wheezing due to rhinovirus infection and due to laryngitis on an allergic basis”. Pag 3 lines 98-99 we add the sentences: General conditions were good, with normal vital signs and laboratory parameters, in-cluding cardiac biomarkers (Troponin I and NT-proBNP) constantly in the normal range. Physical examination including cardiac auscultation was normal, and she had no evi-dence of JVP elevation, peripheral edema or hepatomegaly”.

  1. Mutations in FLNC gene have been associated with human hypertrophic cardiomyopathy, restrictive cardiomyopathy, and a higher incidence of sudden cardiac death. At present, this mutation can only be an evidence to support the diagnosis, not the confirmation of the diagnosis. Regarding what the authors mention “The test is strongly recommended in patients with hereditary cardiomyopathies, in this way the test can also have a predictive role, in ascertaining which family members are most at risk.” Is it possible to provide evidence for this claim such as what is the incidence of RCM in the presence of FLNC mutations?

Answer:  Thank you for the comments.  We added in the text a brief paragraph to provide evidence about the incidence of RCM in the presence of FLNC mutations:

Page 2 lines 70-81 we add the sentences The reported estimated prevalence of FLNC variants in RCM patients has been poorly defined, ranging from 2.2% [16] to 8% [13]. In the HGMD Professional database (2022.1) 156 disease-causing variants in FLNC are reported, associated with an isolated cardiomyopathy (HCM, DCM, RCM, LVNC) and with a myopathy phenotype; 11/156 FLNC variants (7%) were associated with RCM. Because of the rarity of RCM due to FLNC variants, its influence on prognosis and therapy is poorly defined compared with other cardiomyopathies, particularly in children [14]. FLNC missense variants can cause early onset RCM associated with variable degrees of a skeletal myofibrillar myopathy, combined with mild CK elevation and supraventricular arrhythmias. Differently from missense mutation, FLNC truncating variants have been found in patients with an arrhythmogenic/DCM phenotype characterized by a high risk of life-threatening ventricular arrhythmias [15,16]”.

We add the following citations:  [16] Ortiz-Genga MF., Cuenca S., Dal Ferro M., Zorio E., Salgado-Aranda R., Climent V., Padrón-Barthe L., Duro-Aguado I., Jiménez-Jáimez J., Hidalgo-Olivares VM., García-Campo E., Lanzillo C., Suárez-Mier MP., Yonath H., Marcos-Alonso S., Ochoa JP, Santomé JL, García-Giustiniani D, Rodríguez-Garrido JL, Domínguez F, Merlo M, Palomino J, Peña ML, Tru-jillo JP., Martín-Vila A., Stolfo D., Molina P., Lara-Pezzi E., Calvo-Iglesias FE., Nof E., Calò L., Barriales-Villa R., Gime-no-Blanes JR., Arad M., García-Pavía P., Monserrat L. Truncating FLNC Mutations Are Associated With High-Risk Di-lated and Arrhythmogenic Cardiomyopathies. J Am Coll Cardiol. 2016 Dec 6;68(22):2440-2451.

Reviewer 2 Report

The article can be improved to make it more scientifically sound and easy for everyone to understand, even those not working in genomics. 

Author Response

We would like to thank Reviewer #2 for the comments.

Reviewer 3 Report

Good case report on established mutation in FLNC detected on clinical exome analysis emphasising the poor outlook for these paediatric cases and need to refer for early transplantation. 

Author Response

We appreciate all the comments from the reviewers.